# Farmer preference for macadamia varieties and constraints to production in Malawi

**Emmanuel Junior Zuza**[1,2]*, **Yoseph N. Araya**[2], **Kadmiel Maseyk**[2], **Shonil Bhagwat**[3], **Rick L. Brandenburg**[4], **Andrew Emmott**[5], **Will Rawes**[5], **Patrick Phiri**[6], **Ken Mkengala**[7], **Edwin Kenamu**[8]

1 School of Agricultural Science and Practice, Royal Agricultural University, Cirencester, United Kingdom, 2 School of Environment, Earth and Ecosystem Sciences, The Open University, Milton Keynes, United Kingdom, 3 School of Social Sciences and Global Studies, The Open University, Milton Keynes, United Kingdom, 4 Department of Entomology and Pathology, North Carolina State University, Raleigh, North Carolina, United States of America, 5 The Neno Macadamia Trust, Bedford, United Kingdom, 6 School of Environmental Design and Horticulture, Writtle University College, Chelmsford, United Kingdom, 7 Highlands Macadamia Cooperative Union Limited, Ntchisi, Malawi, 8 Department of Agricultural Economics and Rural Development, Georg-August-Universität, Göttingen, Germany

* Emmanuel.Zuza@rau.ac.uk

**Data Availability Statement:** The minimal data set is contained on the figshare link (https://figshare. com/articles/dataset/Farmer_preference_for_

## Abstract

Macadamia nuts constitute a vital component of both nutrition and livelihoods for small-holder producers in Malawi. We conducted a comprehensive mixed-methods study, combining qualitative and quantitative analyses, to explore varietal preferences and production challenges among these farmers. Leveraging cross-sectional data from 144 members of the Highlands Macadamia Cooperative Union Limited, our study underscores several significant findings. Our findings reveal that the majority of smallholder macadamia farmers (62%) are aged over 50, with farming as their primary occupation. Varied preferences are driven by yield-related traits, including high yield potential (38%), nut quality (29%), and extended flowering patterns (15%). Among the macadamia varieties, the top five choices, grown by over half of the farmers, include HAES 660 (18%), 800 (10%), 791 (9%), 816 (8%), and 246 (7%). Additionally, our study identifies five primary constraints faced by smallholder macadamia farmers: insect pests (81%), diseases (34%), limited market access (33%), wind damage (25%), and inadequate agricultural advisory services (17%). Based on these findings, we propose two policy recommendations to enhance smallholder macadamia production and productivity in Malawi and other regions. Specifically, we advocate for informed breeding programs that align with farmer preferences to promote greater adoption of macadamia varieties. Additionally, we emphasize the crucial role of the Malawian government in the macadamia value chain, suggesting active participation in providing extension services and marketing support, akin to its support for other cash crops.

## Introduction

Horticulture is vital in providing food and income for many households in Malawi [1, 2]. Despite utilizing only a small fraction of the country's arable land ($\leq$ 5%), the horticulture

macadamia_varieties_and_constraints_to_production_in_Malawi_Raw_data/24866670).

**Funding:** The corresponding author received funding from the UK Research and Innovation - Global Challenges Research Fund to pursue their PhD studies at the Open University. Additionally, the corresponding author received training from Earth Watch Community Science Camp (NERC-UKRI Grant no. NE/S017437/1). Furthermore, the corresponding author received travel funding from North Carolina Agricultural and Technical State University, through Feed the Future, Peanut Innovation Lab.

**Competing interests:** No cempeting interests.

industry has enormous potential to contribute to both national food security and economic growth [3]. Malawi is renowned for its diverse fruit crops, including bananas, citrus species, coffee, macadamia nuts, and tea. Bananas and citrus are particularly important, constituting a staple part of the diet in rural and urban areas, with approximately 95% of their production consumed domestically [4, 5]. Additionally, coffee and tea are major products consumed locally and exported [6, 7].

Macadamia (Proteaceae) is one of the world's most profitable crops, and its demand is increasing worldwide. Malawi has a thriving macadamia industry and is regarded as a premium origin worldwide [8, 9]. The crop is well-established and grown on commercial estates and smallholder landholdings nationwide. Because the crop's returns are higher (~$14 to 15 kg$^{-1}$) than other cash crops, the country's macadamia industry is rapidly expanding, with farms previously used for coffee, tea, and tobacco production being converted or diversified to macadamia production [10, 11]. Therefore, macadamia is an important crop among smallholder farmers as a food source and cash crop.

However, macadamia yields among smallholder farmers in Malawi have been reported to be significantly below the optimal level, with an average of 200 kg per hectare per year compared to the ideal 1500 kg ha$^{-1}$ year$^{-1}$ [12, 13]. Several factors contribute to this disparity, including suboptimal management practices, adverse weather conditions, limited access to high-quality macadamia tree seedlings, soil nutrient deficiencies, and insect pests and diseases. Previous research has suggested that new generation (refers to distinct period of varietal development) macadamia varieties (NGMV) with enhanced yield potential and improved resilience to pests and diseases can elevate smallholder productivity [14, 15]. These NGMVs are higher yielding ($\geq$ 45 kg tree$^{-1}$ year$^{-1}$) than old generation varieties (OGMVs), which yield less than 30 kg tree$^{-1}$ year$^{-1}$ [16–19].

Despite the availability of NGMVs, their adoption by smallholders has been slow [11, 20]. This is attributed to various socioeconomic factors such as limited awareness, misconceptions about the associated costs and benefits, and farmer preferences. Nonetheless, it remains imperative for smallholders to embrace these NGMVs, as they hold the key to the long-term growth and sustainability of the macadamia industry in Malawi. Furthermore, in response to market demands, macadamia processors in Malawi are actively promoting the adoption of NGMVs, underlining the critical importance of this transition in the macadamia sector.

Adopting new agricultural technologies, including varieties, is rarely immediate, particularly for high-value perennials [21, 22]. New agricultural technologies are often associated with risks and uncertainties regarding the appropriate application, scalability, environmental compatibility, and, most importantly, farmers' perceptions and expectations [23–26]. Subsequently, when working in agricultural systems such as those found in Malawi, it is vital to identify the factors that may influence the adoption of new technologies. Studies have shown that household characteristics, perceived benefits, market availability, input costs, and long versus short-term benefits influence the adoption of an agriculture technology [22, 27, 28].

Participatory Rural Appraisal (PRA) can help overcome challenges that prevent farmers from adopting new agricultural technologies [29, 30]. By incorporating farmer knowledge and perspectives into the planning and management of research development initiatives, PRA increases the likelihood of farmers adopting newly developed agriculture technologies [31, 32]. This approach has been successfully applied in different countries, leading to the development of diverse groundnut varieties with desirable attributes for Malawian farmers [33, 34] and improved variety adoption of potatoes in Ghana, Nigeria, and Uganda [22, 30]. Therefore, this study uses the PRA to examine smallholder farmers' preferences for macadamia traits and cultivars in Malawi. The specific objectives of this study are as follows:

1. Identify the determinants and preferred macadamia traits and varieties among smallholder macadamia farmers.

2. Analyse the challenges and opportunities experienced by smallholder macadamia farmers.

## Materials and methods

This section discusses the following: study sites, research design, target population, sample size, sampling procedure, data collection, and data analysis methods.

### Study location

Our study focuses on smallholder macadamia farmers who are members of the seven primary Highlands Macadamia Cooperative Union Limited cooperatives (HIMACUL). HIMACUL is a Malawian smallholder-owned macadamia cooperative and works with seven district level primary cooperatives (Chikwatula, Malomo, Tithandizane and Mphaza, Nachisaka, Neno, and Mwanza). It operates in Dowa, Mwanza, Neno, and Ntchisi districts (Fig 1), but previously included Rumphi district. The main focus of HIMACUL is the promotion of macadamia agro-forestry and nut trading with its member farmers [35]. The selection of HIMACUL farmers as

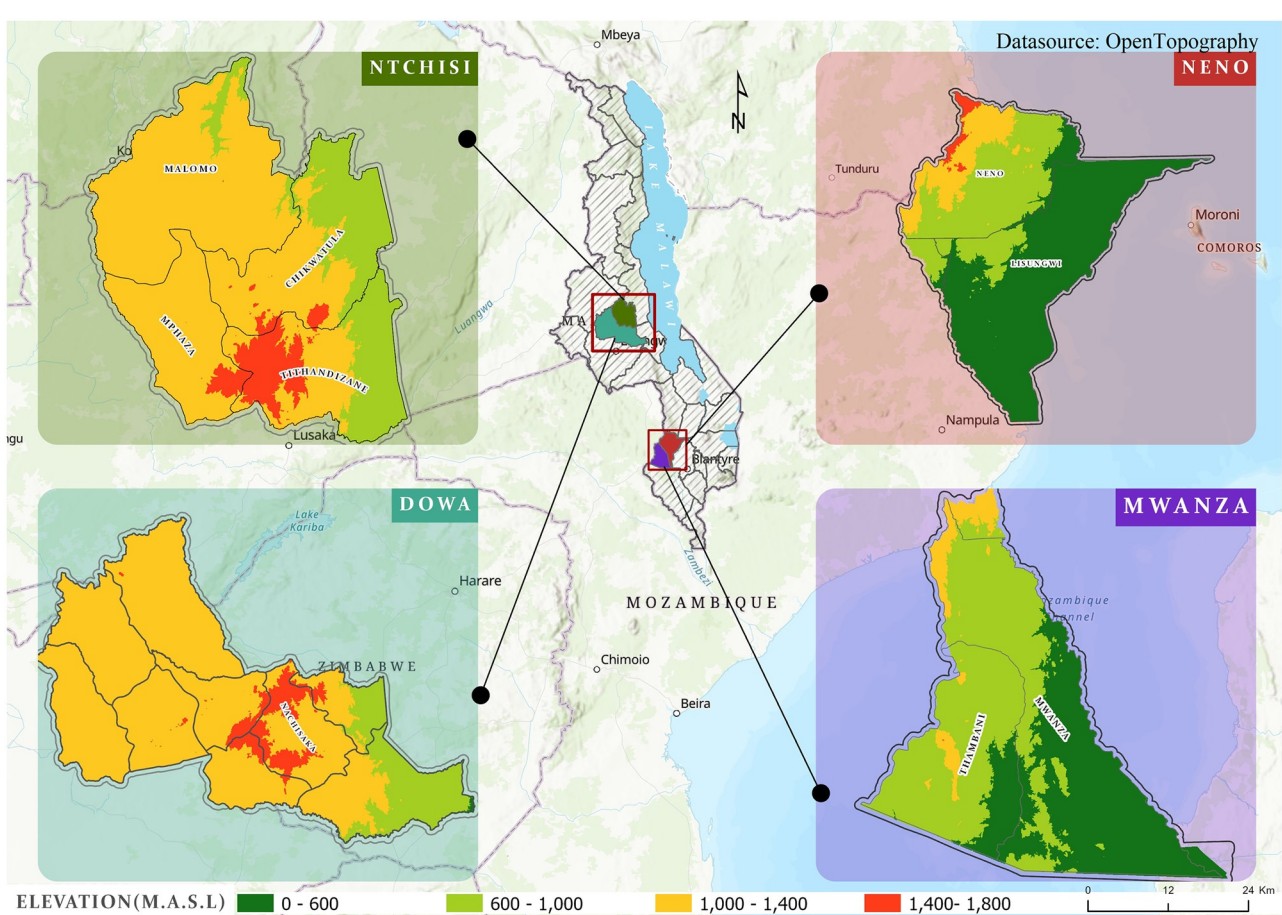

**Fig 1. Map of Malawi showing the study areas and associated elevation.** Source: Authors.

**Table 1. Annual average climate conditions of the study areas.**

| District | Cooperative | Precipitation (mm) | Annual min temp (˚C) | Annual max temp (˚C) |
|----------|-------------|--------------------|-----------------------|-----------------------|
| Ntchisi | Chikwatula | 1200–1500 | 10 | 30 |
| | Malomo | 800–1200 | 11 | 35 |
| | Tithandizane | 1200–1800 | 8 | 25 |
| | Mphaza | 1000–1500 | 12 | 32 |
| Dowa | Nachisaka | 800–1500 | 10 | 32 |
| Neno | Neno | 800–1200 | 15 | 35 |
| Mwanza | Mwanza | 600–1000 | 15 | 40 |

the focus of our study was deliberate, as they serve as a representative cross-section of small-holder macadamia producers in Malawi [36].

Dowa district covers an area of 3041 km$^2$ with altitudes varying between 700 to 1500 m.a.s.l. The district has seven Extension Planning Areas (EPAs), with macadamia cultivation concentrated in Nachisaka EPA. Extension planning areas are geographic subdivisions or regions established for agricultural extension and development planning [37–39]. The district experiences an annual temperature range of 10–32˚C and annual precipitation from 800–1500 mm (Table 1), influenced by elevation. Ntchisi district, covering 1655 km$^2$ with altitudes ranging from 900 to over 2000 m.a.s.l., exhibits diverse agroecology across the HIMACUL cooperatives. For example, Tithandizane cooperative spans from hot and dry low-lying areas on the Rift Valley escarpment to cool and wet regions on the Rift Valley ridge. Precipitation varies among the cooperatives, with Malomo receiving lower levels (800–1200 mm) than Chikwatula and Tithandizane (1200–1500 mm). The district experiences an average annual temperature range of 8˚C in winter to 35˚C in summer.

Mwanza district covers an area of about 826 km$^2$. It has elevations ranging from 600 to 1200 m.a.s.l. The average annual temperatures vary from 15˚C in high and 40˚C in lower altitudes. Annual precipitation varies between 600 and 1000 mm. Macadamia is grown in Mwanza EPA. Neno district, covering approximately 1469 km$^2$, has a varied landscape, with hilly areas exceeding 2000 m.a.s.l. in the Kirk range and flatter areas in Lisungwi and the Shire Valley (800 and 1000 m.a.s.l.), creating a stark contrast in agroecological conditions within the district. The district has two EPAs, Lisungwi and Neno, known collectively as Neno cooperative. The temperature ranges from 15˚C in high-altitude areas to 35˚C in the Shire Valley's low-lying areas, with an annual average precipitation of 800–1200 mm.

## Sampling technique and data collection

**Ethics.** The Human Research Ethics Committee (HREC) of the Open University, United Kingdom, provided ethical approval for this study (HREC/3306/Zuza). Participants provided formal consent verbally and signed a consent form after being briefed on the study's objectives in their local language. The study participants were assured that their identities would remain confidential and anonymous.

**Qualitative data.** Qualitative data were collected using Focus Group Discussions (FDGs). A purposive sampling strategy was used to sample the participants in our study. The farmers were selected based on their active involvement in trading macadamia nuts with HIMACUL, ownership of at least one hundred macadamia trees, and having at least two consecutive harvests from their macadamia orchards. A total of seven FDGs were conducted across the seven HIMACUL cooperatives. The researchers ensured that the FGDs had a representative number of males and females, including the youth. The FDGs were conducted in two phases: gender-

specific groups (men only and female only) and mixed groups. The separation aimed at addressing cultural norms that may make it difficult for women to express themselves in the presence of men or village elders. However, during the mixed FGDs, women were free to speak in the presence of men. The participants provided their opinions on their preferred macadamia traits and cultivars. The FDGs were conducted from 01[st] September 2019 to 25[th] November 2019.

**Quantitative data.** Primary data (survey) was collected by research assistants from the seven primary HIMACUL cooperative members within the period highlighted in Section 2.2.2. The questionnaire was structured to address specific objectives and was based on methods for examining agricultural problems and assessing farmers' knowledge, perceptions, and practices [40]. A total of 144 participants were purposively sampled during the survey. The survey employed a combination of open and closed-ended questions to collect information on demographic characteristics, preferred macadamia traits and cultivars, and production constraints.

## Model formulation

**Theoretical model for farmer preference for macadamia varieties.** Farmer preference for macadamia traits and cultivars in Malawi can best be assessed by combining the Random Utility theory [41] with Lancaster's theory of consumer demand [42]. From the random utility theory, a farmer faces a choice set $C$ that has $j$ macadamia varieties. The farmer's task is to select cultivars that maximise their utility function $U$. To select the utility maximising variety, the farmer's decision rule is to compare $U_1$, $U_2$, . . .$U_j$ and select the outcome that gives the maximum utility. Given the utility functions, farmer $i$ only selects a variety $j$ over cultivar $k$ in time $t$ if and only if:

$$U_{ijt} > U_{ikt}, \forall j \neq k \, \forall k \in J \tag{1}$$

Each farmer is assumed to have a latent utility function $U_{ijt}^*$ that cannot be observed directly. What is directly observable to the researcher is the choice $U_{ijt}$, where $U_{ijt}^* = 1$ *if* $\max(U_{i1t}^*, U_{i2t}^* \ldots, U_{ijt}^*)$, and 0 if $J$ otherwise. Note for all varieties in set with elements $j$, $k$, the farmer chooses a variety that gives them the most utility/profit when they compare $j$ and $k$.

From the Lancasterian theory [42], it can be argued that farmers do not derive utility directly from a variety in its "totality." Instead, a farmer's selection of a variety is based on the variety's fundamental traits (characteristics). The assumption is that the farmer is interested in the varietal characteristics, not the variety's entire form. Even if all other macadamia characteristics are identical, a farmer may benefit more from cultivating a disease-resistant variety than a high-yielding one. In this regard, it is not the variety as a whole that provides value to the farmer but rather its characteristics, in this case, disease resistance. To analyze farmer preference for macadamia varieties more thoroughly, one must pay particular attention to the varietal traits and how they influence the farmer's choice. However, when smallholder macadamia farmers first planted their macadamia trees, there were few options for planting materials [10]. And farmers knew little to nothing about the traits of the varieties. Choice in this context must consider the farmers' "preferred variety" based on their collective and individual experiences.

A model that only accounts for macadamia traits cannot adequately explain farmer preferences for macadamia varieties. In essence, the characteristics of the farmer and the variety influence the probability that a farmer will prefer one variety over others. Therefore, any model that attempts to quantify the probability of a variety being chosen as the preferred option should also incorporate farmer and farm attributes. Agroecology, age and sex of the

farmer, access to agricultural advisory services, education levels, and household size have significantly influenced farmer preferences for various crop varieties [43–45].

There is no reason to believe these factors do not influence farmers' preference for macadamia varieties. Therefore, any model that seeks to understand farmer preference for macadamia varieties should incorporate the aforementioned farmer and farm attributes to quantify how they condition the probabilities for the various varieties to be chosen as the preferred ones. However, not all factors influencing farmer preference for macadamia traits are known to the researchers. Some of these factors are unobservable, allowing the farmers' utility function from cultivars to be expressed as follows:

$$U = \beta\prime x + \varepsilon \tag{2}$$

With $\beta$' being the marginal utilities of the various attributes, $x$ being the observed factors, and $\varepsilon$ being an error term that accounts for the unobserved factors assumed to be random. The statistical distribution of the error term ($\varepsilon$) in Eq 2 determines the distribution of the probability that a farmer will choose any of the $j$ macadamia varieties in Eq 1.

The multiple response nature of the farmer's choice task presents a statistical complexity that the researcher must explicitly account for. Given that the farmer can select up to five varieties based on their characteristics, the probability of selecting one variety and another can be determined jointly. Consequently, the distribution of error terms for the $j$ cultivars can be jointly normal. In such a case, proper analysis of farmer preferences requires a multivariate framework of the Seemingly Unrelated Regression (SUR) type, in which the interrelationships between the $j$ cultivars are explicitly jointly modelled [46]. Given their characteristics, it is also possible that some varieties can serve as substitutes. Therefore, it is crucial to evaluate farmer preferences within a multivariate framework considering the joint distribution of preferences to identify such interrelationships between varieties.

A further complication of the model is the discrete outcome nature of the dependent variables, which are binary indicators of whether or not a farmer selects cultivar $j$ as one of their preferences. The binary nature of the dependent variables necessitates additional assumptions regarding the error term distribution in Eq 2. Since the outcome variables are not continuous, the SUR cannot accurately estimate the model's parameters. Subsequently, a multivariate model with binary outcomes is the most suitable for assessing farmer preferences for macadamia varieties. Continuing with the assumption of multivariate normal distribution of the errors for the $j$ varieties (Eq 2) can be operationalized as follows using a multivariate probit model:

$$y* = \beta' x + \varepsilon_i \tag{3}$$

$$y_i = 1 \text{ if } \beta_i' x + \varepsilon_i > 0, \text{ otherwise } y_i = 0 \text{ if } \beta_i' x + \varepsilon_i \leq 0$$

Where: $y^*$ is the binary outcome, and the other variables and parameters are as previously defined. Notice that the error terms are assumed to be multivariate, normally distributed with a mean zero, a variance of one, and a square simultaneous correlation matrix [47]. One desirable feature of this multivariate model is that it collapses to $j$ univariate probit models when the multivariate normal distribution assumption does not hold, still allowing for empirical estimation of the farmer's preference for the individual varieties.

**Empirical model for farmer preference for macadamia varieties.** Implementing the theoretical model above (Eq 3) requires a substantial variation in the dependent and independent outcomes, preferably from a large enough sample size. However, this study's sample does not allow empirical estimation of the model (preference models for more than 10 varieties require

at least 3000 individuals for empirical analysis [22, 48]. Therefore, empirical estimation of the model becomes difficult as it cannot converge regardless of how it is specified. However, the study ran one probit that explains farmer preference for the variety HAES 660. All data analysis for the econometric model was conducted in STATA® 14.2. The model is specified empirically as follows:

$$y_i = \beta_0 + \beta_1 sex + \beta_2 age + \beta_3 cooperative + \beta_4 education + \beta_5 landsize + \beta_6 droughtresistanse$$
$$+ \beta_7 nutsize + \beta_8 yield + \beta_9 windresistance + \varepsilon_i$$

Where:

$y_i$ = Dependent binary variable (farmer "i" chooses variety HAES 660 = 1, or otherwise = 0)
$\beta_n$ = Parameters estimated, i.e., theorized to affect the choice of the macadamia variety
$\varepsilon_i$ = Error term assumed to be normally distributed.
All data analysis for the econometric model was conducted in STATA® 14.2.

## Results

### Demographic characteristics

Among HIMACUL cooperatives, the average age of a farmer was 50 years in 2019 (Table 2), indicating an ageing population and the need to encourage the youth to engage in macadamia production. A younger farmer (aged 30) indicated:

"There is little youth involvement in macadamia farming. This is mainly because of the lack of land and the waiting time to start harvesting the nuts. However, with agroforestry training, I have seen many youthful farmers starting to grow macadamia trees."

(Male young farmer, Mwanza cooperative).

We found that 58% of the study participants were males and the remainder being females. Interestingly, more than 90% of the respondents reported owning their macadamia orchards in partnership with their spouses. A male farmer commented:

"Macadamia farming is a serious venture that requires high initial capital investment, proper management of tree seedlings, and patience. My wife and I have equal ownership of our macadamia farm as we have invested money and our entire life looking after the trees."

(Male farmer, Tithandizane cooperative).

Regarding education levels, our results revealed that about 53% of HIMACUL farmers have completed the Malawi Primary School Certificate of Education, while 32% have completed secondary or university education. Approximately 80% of the respondents were married, with the remaining being single (4%), divorced (3%), or widowed (13%). Most HIMACUL farmers (89%) consider farming their primary occupation, implying that agriculture is their main source of income (Table 2). Nevertheless, about 6% of the farmers are engaged in alternative ventures, such as operating motorcycle taxis.

### Macadamia varieties common among HIMACUL farmers

Smallholder macadamia farmers in Malawi commonly cultivate a diverse range of 18 macadamia varieties (Fig 2). Notably, HAES 660 was the most popularly grown variety, accounting for 49% of the total, followed by HAES 800 (26%), 791 (19%), 333 (13%), and 246 (12%). On

**Table 2. Demographic characteristics by cooperative.**

| Variable | Cooperative name | | | | | | | |
|---|---|---|---|---|---|---|---|---|
| | Chikwatula (n = 20) | Malomo (n = 19) | Mphaza (n = 20) | Tithandizane (n = 24) | Mwanza (n = 24) | Nachisaka (n = 18) | Neno (n = 24) | Overall (N = 144) |
| **Gender (%)** | | | | | | | | |
| **Male** | 6.3 | 4.9 | 6.3 | 2.1 | 10.4 | 2.8 | 9.0 | 41.7 |
| **Female** | 7.6 | 8.3 | 7.6 | 13.9 | 6.3 | 9.7 | 4.9 | 58.3 |
| **Age (%)** | | | | | | | | |
| **18–30** | 0.7 | 0.0 | 0.7 | 0.7 | 1.4 | 0.0 | 1.4 | 4.9 |
| **31–40** | 0.7 | 0.7 | 1.4 | 0.0 | 0.7 | 0.7 | 0.7 | 6.9 |
| **41–50** | 2.8 | 3.5 | 3.5 | 4.2 | 6.9 | 4.1 | 1.4 | 4.9 |
| **51–60** | 4.9 | 2.8 | 4.2 | 6.3 | 2.8 | 3.5 | 2.8 | 27.1 |
| **61–70** | 3.5 | 2.8 | 2.8 | 2.8 | 2.1 | 2.1 | 3.5 | 19.4 |
| **≥71** | 1.4 | 3.5 | 1.4 | 2.1 | 2.8 | 2.1 | 2.1 | 15.3 |
| **Marital status (%)** | | | | | | | | |
| **Single** | 0.7 | 0.0 | 0.7 | 0.0 | 0.7 | 0.0 | 1.4 | 3.5 |
| **Married living with spouse** | 9.7 | 13.2 | 10.4 | 14.6 | 12.5 | 11.8 | 7.6 | 79.9 |
| **Married living without spouse** | 0.0 | 0.0 | 0.0 | 0.0 | 0.0 | 0.7 | 0.0 | 0.7 |
| **Divorced** | 0.0 | 0.0 | 1.4 | 0.0 | 0.7 | 0.0 | 0.7 | 2.8 |
| **Widowed** | 3.5 | 0.0 | 1.4 | 1.4 | 2.8 | 0.0 | 4.2 | 13.2 |
| **Education (%)** | | | | | | | | |
| **No formal education** | 1.4 | 2.1 | 2.8 | 4.2 | 3.5 | 0.7 | 0.7 | 15.3 |
| **Primary (1–8)** | 6.3 | 6.9 | 8.3 | 9.7 | 8.3 | 6.9 | 6.3 | 52.8 |
| **Secondary** | 2.8 | 4.2 | 2.8 | 2.1 | 4.2 | 4.9 | 6.3 | 27.1 |
| **Tertiary (College & University** | 3.5 | 0.0 | 0.0 | 00 | 0.7 | 0.0 | 0.7 | 4.9 |
| **Occupation (%)** | | | | | | | | |
| **Agriculture self-employed** | 12.5 | 12.5 | 13.2 | 14.6 | 13.2 | 11.1 | 11.8 | 88.9 |
| **Informal unskilled labour** | 0.7 | 0.0 | 0.0 | 0.0 | 1.4 | 0.0 | 0.0 | 2.1 |
| **Domestic worker** | 0.0 | 0.0 | 0.0 | 0.0 | 0.7 | 0.0 | 0.0 | 0.7 |
| **Pensioner** | 0.7 | 0.0 | 0.0 | 0.0 | 0.0 | 0.7 | 0.7 | 2.1 |
| **SME owner** | 0.0 | 0.7 | 0.7 | 1.4 | 1.4 | 0.7 | 1.4 | 6.3 |

Note: Brighter colours indicate higher percentages.

average, these farmers manage an orchard with approximately seven to eight varieties. Interestingly, our study shows that some farmers have already adopted NGMVs, including HAES 772 (12%), 816 (11%), and 741 (11%). However, our findings revealed that wider adoption of certain newer varieties introduced by commercial estate producers in 2010, such as Beaumont, Daddow, and MCT1, has not yet been observed among HIMACUL farmers. An expert provided this insightful remark during the interviews:

> "Commercial estate producers in Malawi have imported a range of new generation varieties, including MCT1 protected by Plant Breeders Rights. As a result, such varieties are currently not available to smallholder farmers due to propagation agreements."

(HIMACUL staff member, Chikwatula cooperative).

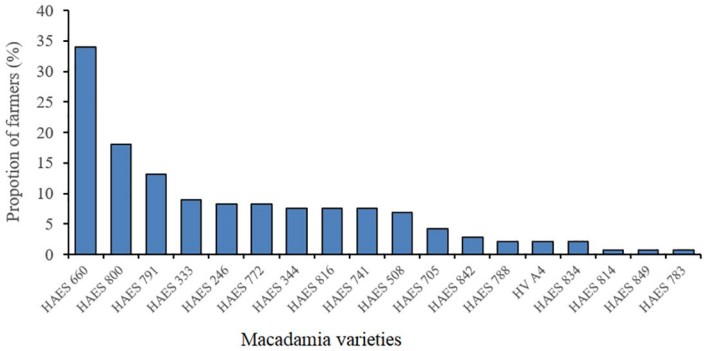

**Fig 2. Macadamia varieties common among smallholder farmers in the study areas.**

## Drivers of macadamia variety preference

Farmer perceptions are a key factor in determining the overall acceptability of macadamia varieties. Fig 3 shows that smallholder macadamia farmers use a combination of similar criteria to select desirable traits for preference of macadamia varieties. The most important characteristics that influence farmer choices for a macadamia variety are high yielding potential (38%), nut quality (29%), and extended flowering period (15%). However, for the HAES 660 cultivar, rootstock and price benefits are the most important considerations for farmers. A farmer indicated:

> "We all love the HAES 791 variety because it flowers 2 to 3 times a year, resulting in higher yields. The only drawback of this variety is its lack of resistance to stinkbug damage."

(Male farmer, Nachisaka cooperative).

The findings obtained from our probit model indicate notable variations in the preferred characteristics linked to different macadamia varieties among smallholder farmers. Our study

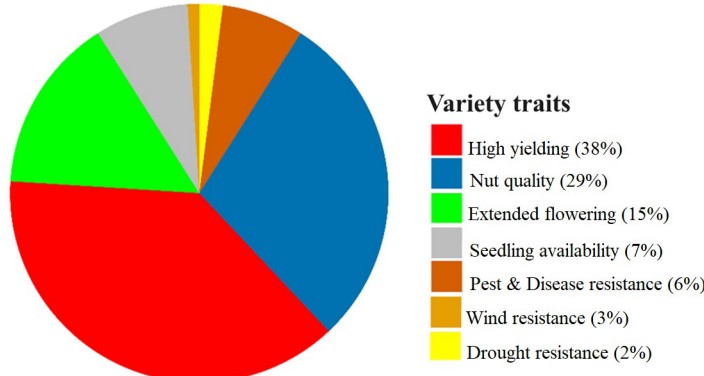

**Fig 3. Proportion of HIMACUL members stating a preference for a given macadamia variety trait.**

**Table 3. Summary statistics for preferred macadamia variety traits.**

| Cultivar attribute | Chikwatula | Malomo | Mphaza | Mwanza | Nachisaka | Neno | Tithandizane | Prob > F |
|---|---|---|---|---|---|---|---|---|
| High yield | 0.600 | 0.263 | 0.300 | 0.167 | 0.389 | 0.550 | 0.478 | 0.032** |
| | (0.503) | (0.452) | (0.470) | (0.381) | (0.502) | (0.510) | (0.511) | |
| Nut quality | 0.600 | 0.368 | 0.100 | 0.167 | 0.222 | 0.300 | 0.348 | 0.014*** |
| | (0.503) | (0.496) | (0.308) | (0.381) | (0.428) | (0.470) | (0.487) | |
| Flowers all year round | 0.250 | 0.105 | 0.100 | 0.125 | 0.167 | 0.250 | 0.130 | NS |
| | (0.444) | (0.315) | (0.308) | (0.338) | (0.383) | (0.444) | (0.344) | |
| Seedling availability | 0.150 | 0.053 | 0.050 | 0.042 | 0.167 | 0.250 | 0.000 | 0.081* |
| | (0.366) | (0.229) | (0.224) | (0.204) | (0.383) | (0.444) | (0.000) | |
| Pest and disease resistance | 0.100 | 0.105 | 0.050 | 0.000 | 0.111 | 0.100 | 0.130 | NS |
| | (0.308) | (0.315) | (0.224) | (0.000) | (0.323) | (0.308) | (0.344) | |
| Drought resistant | 0.000 | 0.000 | 0.000 | 0.042 | 0.000 | 0.000 | 0.000 | NS |
| | (0.000) | (0.000) | (0.000) | (0.204) | (0.000) | (0.000) | (0.000) | |
| Wind resistant | 0.100 | 0.000 | 0.100 | 0.042 | 0.000 | 0.000 | 0.000 | NS |
| | (0.308) | (0.000) | (0.308) | (0.204) | (0.000) | (0.000) | (0.000) | |

Note: Standard deviations are in parentheses.

*** Significant at P ≤ 0.001,

** Significant at P ≤ 0.05,

*Significant at P ≤ 0.1 and NS = Not significant.

shows a significant influence of high yielding ($P = 0.032$) and nut quality ($P = 0.014$) traits on selecting macadamia varieties among the sampled smallholders. The greatest preference for high yielding varieties is observed in Chikwatula (60%) and Neno (55%) cooperatives (Table 3). On the other hand, farmers belonging to Malomo (36.8%), Neno (30%), and Tithandizane (34.8%) cooperatives prioritize varieties that have higher nut quality in terms of weight and size.

Furthermore, our study underscores the important role played by the geographical location of HIMACUL cooperatives in shaping varietal choices among farmers. For instance, farmers in Chikwatula (31%), Mphaza (31%), and Mwanza (20%) cooperatives prefer varieties that are resistant to wind due to their specific location. Farmers reported that these areas experience heavy seasonal winds, leading to significant flower and nut drops and lower crop yields. Additionally, the farmers suggested that, apart from temperature and precipitation, wind plays a crucial role in determining the climate suitability of macadamia cultivation in these areas.

## Macadamia variety preference

The results of the FGDs identified a range of macadamia varieties preferred by smallholder macadamia farmers who are members of HIMACUL. Participants ranked their preferred macadamia varieties. The findings show that these farmers prefer the HAES 660 variety over other available varieties. The NGMVs HAES 800 and 791 were ranked second and third preferred varieties, respectively (Table 4). The least preferred cultivars include HAES 783, 788, 834, and 849.

Our study reveals distinct preferences for specific macadamia varieties among individual cooperatives. Within Chikwatula cooperative, HAES 246 (46%) is the most preferred variety, while HAES 772 (5%) ranks as the least preferred variety (Fig 4). Conversely, in Malomo cooperative, HAES 791 (35%) is the most popular, while HAES 741 is the least popular. Meanwhile, Mphaza farmers strongly prefer HAES 741 (19%), with HAES 333 (4%) as the least preferred

**Table 4. Macadamia variety preference by ranking.**

| Variety | Rank |
|---|---|
| HAES 660 | 1 |
| HAES 800 | 2 |
| HAES 791 | 3 |
| HAES 816 | 4 |
| HAES 772 | 5 |
| HAES 246 | 6 |
| HAES 741 | 7 |
| HAES 333 | 8 |
| HAES 344 | 9 |
| HAES 508 | 10 |

variety. In the Mwanza cooperative, farmers expressed the greatest preference for cultivars HAES 741 and 800 (24%0. Among Nachisaka cooperative farmers, HAES 791 (30%) is the preferred variety. The majority of farmers in Neno cooperative prefer HAES 660 (70%), with HAES 772 being the least favoured. In Tithandizane cooperative, the most preferred cultivar is HAES 333 (24%), and the least preferred cultivar is HAES 842.

## Determinants of preference for H660 variety: Econometric model results

In this section we examine factors influencing the preference for the HAES 660 variety. Our analysis reveals that cooperative membership significantly influences the preference for HAES 660. More specifically, the results obtained from the probit model in Table 5 show that farmers affiliated with Malomo and Neno cooperatives are 34.5% (marginal effect of 0.345) and 50% (marginal effect of 0.501) more likely to prefer the HAES 660 variety compared to other varieties. However, farmers from Mphaza cooperative are less likely (–0.0129) to prefer HAES 600. These findings shed light on the role of cooperative membership in shaping varietal preferences among smallholder macadamia farmers in Malawi. A male farmer from Mphaza cooperative reported:

> "Here in Mphaza, we have access to the HAES 741 variety, which produces more yields than other varieties. Moreover, Sable Farming has been distributing seedlings for the variety and provide a ready market as well. Therefore, our inclination to adopt this variety stems from its accessibility and promising outlook."

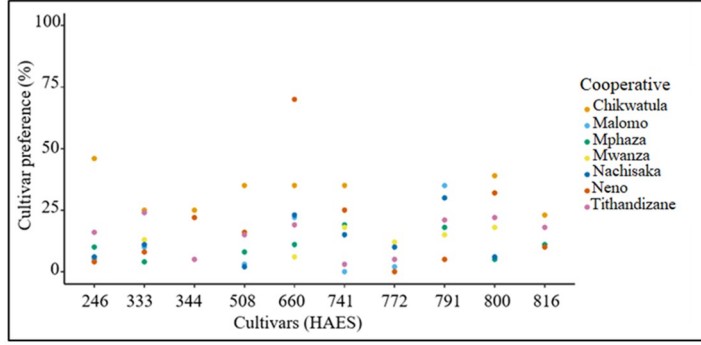

**Fig 4. Macadamia preference at the individual cooperative level.**

**Table 5. Probit analysis of determinants of HAES 660 preference.**

| Variable | Coefficient | Marginal effects |
|---|---|---|
| **Cooperative** | | |
| Malomo | 2.842 | 0.345* |
| Mphaza | −0.131 | −0.0129 |
| Mwanza | N/A | N/A |
| Nachisaka | 1.332 | 0.157 |
| Neno | 3.906* | 0.501*** |
| Tithandizane | 0.411 | 0.0451 |
| **Sex** | | |
| Male | 1.647 | 0.225** |
| **Education** | | |
| Primary | −0.156 | −0.0229 |
| Secondary | −2.940 | −0.313** |
| Tertiary | −2.627 | −0.289 |
| **Cultivar attributes** | | |
| High yielding in total | 0.275 | 0.0465 |
| Big nuts | −2.370** | −0.412** |
| Flowers all year round | −1.376 | −0.239 |
| Resistant to pests & diseases | −0.203 | −0.0353 |
| Drought resistant | N/A | N/A |
| Easy to find seedlings | 1.023 | 0.178 |
| Wind resistant | N/A | N/A |

*** Significant at $P \leq 0.001$,

** Significant at $P \leq 0.01$,

*Significant at $P \leq 0.05$

On gender, the probit model results indicate that being male is associated with a higher likelihood of preferring the HAES 660 variety. Specifically, for the model probit, the probability ratio on "sex" is statistically significant ($P \leq 0.01$), and the value of the marginal effect is 0.225, meaning that being male instead of female increases the likelihood of preferring the variety HAES 660 by 22.5% (Table 5). Attesting to this, a farmer pointed out that:

"HAES 660 is a valuable variety for several reasons. Firstly, it fetches a better price from HIMACUL. Additionally, it is commonly used as a rootstock for grafting other macadamia varieties. If you have a nursery, you can generate extra income by selling macadamia tree seedlings. That is why many of us, especially the men in our community, maintain nurseries to boost our earnings."

(Female farmer, Neno cooperative).

The HAES 660 variety is susceptible to damage from insect pests, specifically stinkbugs. As expected, our analysis using model indicates that farmers with higher levels of education demonstrate a greater propensity for not preferring HAES 660. The marginal effects associated with the variable show a decreasing trend as the education level increases. Specifically, the marginal effects are −2.3% for primary education, −31.3% for secondary education (most farmers have only attained this type), and −28.9% for tertiary education.

**Table 6. Challenges faced by smallholder macadamia farmers.**

| Constraint | Category | Frequency | Percentage (%) | Rank |
|---|---|---|---|---|
| Insect pests | Agronomic | 117 | 81 | 1 |
| Diseases | Agronomic | 49 | 34 | 2 |
| Market availability | Socioeconomic, policy | 47 | 33 | 3 |
| Wind | Natural | 37 | 25 | 4 |
| Lack of agricultural extension services | Technical, policy | 24 | 17 | 5 |
| Shortage of seedlings | Socioeconomic, policy | 22 | 15 | 6 |
| Droughts | Natural | 17 | 12 | 7 |
| Lack of labour | Socioeconomic, household | 11 | 8 | 8 |
| Poor transportation | Socioeconomic, policy, household | 10 | 7 | 9 |
| Poor soil fertility | Biophysical, natural, socioeconomic | 9 | 6 | 10 |
| Climate change | Natural | 7 | 4 | 11 |
| Limited land | Socioeconomic | 2 | 1 | 12 |

Note: Multiple responses.

Our model findings reveal distinct preferences among HIMACUL farmers regarding variety traits. Notably, our study indicates that smallholder macadamia farmers are more likely to prefer varieties that produce bigger nuts, have multiple flowering patterns, and are pest and disease resistant compared to HAES 660 (Table 5). More specifically, the odds ratios associated with HAES 660 variety traits, including big nuts, flowering all year round, and resistance to pests and diseases, are statistically significant, and the corresponding marginal effects show negative values, i.e., –41.2%, –23.9%, and –3.5%, respectively. This pattern suggests that smallholder macadamia farmers perceive the HAES 660 variety's smaller nut size and vulnerability to pests and diseases as disadvantages. However, our findings also reveal that farmers are more inclined to prefer the HAES 660 variety due to its seedling availability (17.8%) and high yield potential (4.6%). These preferences appear to be influenced by market considerations, underscoring these varietal traits' pivotal role in shaping farmers' decision-making processes.

## Macadamia production challenges

Among the farmers surveyed, the most prevalent challenges affecting macadamia production are insect pests (81%), diseases (34%), and market availability (33%), as detailed in Table 6. Interestingly, HIMACUL farmers do not regard land availability as a significant constraint that can limit macadamia production. Their explanation points to their access to larger landholding sizes, which effectively mitigates the impact of land availability on their agricultural activities. Our finding underscores the intricate interplay of various factors influencing macadamia production, with potential implications for climate-related considerations in the agricultural landscape.

## Discussion

### Major socioeconomic characteristics of HIMACUL farmers

Our study reveals key insights into the demographic profile of the smallholder macadamia farmers in Malawi. The majority of the sampled smallholder macadamia farmers were male (58%) with 42% being female. This aligns with existing research indicating that male farmers tend to have more control over cash crop management, while females often face limitations in accessing key agricultural resources such as land, inputs, and income [34, 49]. Additionally,

the average age of the participants was 50 years, suggesting that macadamia farmers are not in their active farming years. This affirms early assertions that macadamia farming in Malawi is considered to be a retirement crop [9, 13, 36]. However, approximately 5% of the sampled farmers were aged between 18 and 30. This is an interesting finding, as it indicates an emerging interest among young farmers in macadamia production, potentially leading to expanding their parents' farms or opening new ones. Young farmers' attraction to macadamia farming may be driven by the high market prices of macadamia nuts and the declining popularity of tobacco. Since young farmers are key to the future of agriculture in Malawi [50], their participation in the Macadamia subsector is an encouraging sign of the country's progress toward sustainable and efficient agriculture. In her remarks, the female highlighted the following:

> "I planted *Magede* (a vernacular name for macadamia) because it is my pension and for my children and grandchildren. Once the tree has matured, it will, with little attention to pests and diseases and perhaps some livestock manure, continue growing bigger and giving more nuts yearly".

(Female farmer, Malomo cooperative).

Education is vital in improving economic and agricultural productivity [51]. Educated farmers can better access and use agricultural advisory services and are willing to experiment with modern farming technologies [52]. Our study reveals that over half of the respondents have completed primary school education. Therefore, it can be assumed that if properly trained, they possess the capability to understand and implement good agricultural practices. Furthermore, about 5% of the farmers had attained secondary and university education and are actively involved in macadamia production. This provides an advantage for the Malawian macadamia subsector, as the involvement of educated macadamia farmers can facilitate the adoption of improved agricultural practices, leading to increased productivity.

### Drivers of macadamia variety preferences

Farmers' adoption of new agricultural technologies, such as crop varieties, results from a dynamic interaction between their evaluations of the value of improved varieties and their confidence in varietal's quality [53]. We found that smallholder macadamia farmers in Malawi prefer varieties with multiple desirable attributes, especially yield potential, grain quality (weight and size), and flowering patterns.

Yield potential plays a pivotal role in determining the profitability of macadamia nuts, as farmers have emphasized the significance of yield and nut weight in marketing their macadamia crop. Surprisingly, despite HIMACUL offering higher compensation for HAES 660, which has a smaller kernel size of 23–25 mm (S1 Table), farmers exhibit a higher preference for varieties that yield larger and heavier nuts, such as HAES 246 (28 mm). This preference demonstrates the significance of buyer requirements in influencing farmers' choices for macadamia varieties. Moreover, despite the low-yielding nature of HAES 246 (25 kg tree$^{-1}$ year$^{-1}$), the variety is in high demand among Malawian macadamia processors that target export markets for confectionery macadamia nuts.

Concerning flowering patterns, our qualitative and quantitative results reveal that macadamia farmers prefer varieties with an extended flowering period, such as HAES 791, compared to HAES 660. This preference is due to the correlation between a longer flowering period, extended harvesting times, and subsequently higher yields [54, 55]. This phenomenon is particularly common in high-altitude areas (Chikwatula, Nachisaka, and Tithandizane), where temperatures are cooler, and precipitation is evenly distributed throughout the year, thus

promoting more shoot growth and, as a result, more flowering. In contrast, farmers in low-altitude areas (Malomo, Mwanza, and Neno) reported shorter flowering periods (varieties tend to flower for no more than three months), thus affecting total nut yields. This regional variation helps to explain the differences in macadamia nut yields between Malawi's higher and lower altitude regions, particularly between Malomo and Tithandizane cooperatives [12].

The availability of varietal tree seedlings influences farmers' preferences and, consequently, the adoption of a macadamia variety in Malawi. Farmers expressed concern that most NGMVs are scarce (rootstock) and expensive, leading them to continue cultivating readily available old generation varieties including HAES 660. These findings confirm and, importantly, extend work by Evans [8], who reported that the scion availability of most NGMVs is limited among smallholder macadamia farmers in Malawi, resulting in low adoption levels of the varieties. This is also consistent with the results of our regression analysis, which revealed that farmers prefer the HAES 660 variety because seedlings are readily available. Thus, to increase smallholder farmers' adoption of recommended varieties, particularly NGMVs, it is crucial to have a nursery system to better supply the necessary tree seedlings.

## Macadamia variety preference

Smallholder farmers have specific criteria for selecting crop varieties with preferences varying among communities [22, 56]. Our study findings have shown that HAES 660, 800, 791, 816, and 246 are the most preferred macadamia varieties among HIMACUL farmers based on their ranking. These varieties represent the "core" of macadamia varieties in Malawi [10, 36]. The varieties HAES 246 and 660 have historical significance, as they were recommended in the past (1960s), while varieties HAES 800, 791, and 816 are more recent recommendations from the Malawi Smallholder Development Project (1996–2005). These findings underscore the importance of considering community-specific criteria and preferences when promoting agricultural practices and the adoption of specific crop varieties.

HAES 660 emerged as the most preferred variety among HIMACUL farmers because of its superior rootstock characteristics and a high percentage of grade A kernels. Furthermore, the preference for HAES 660 is driven by HIMACUL, which pays more money for this variety than the other, indicating the potential of leveraging financial incentives to promote the adoption of other recommended varieties, particularly the NGMVs. The second most preferred variety is HAES 800. Farmers cited that the variety is high yielding and resistant to stinkbug damage due to its thick shell. However, its smaller nut size is a notable drawback that deters some farmers. These findings align with Croix & Thindwa [15], who reported a higher percentage of small nuts from HAES 800 than HAES 660. Farmers' preference for HAES 791 is driven by several factors, including its multiple flowering patterns throughout the growing season, quick maturity (initiating nut production after four years), and high yield potential. HAES 816 and 246's popularity is attributed to larger nut size (28 mm) and extended flowering periods.

The second tier of varieties consisting of HAES 333, 344, 772, 741, and 508 appeared less frequently but regularly preferred among HIMACUL farmers. This list presents two generations of varieties: HAES 772 and 741 representing the new generation, while HAES 333, 344, and 508 represent the old generation. Our findings reveal that HAES 333, 741, and 508 are preferred by smallholders in the low-altitude areas of Mphaza, Mwanza, and Neno cooperatives. Despite farmers' preference for HAES 344, Evans [8] reported that the variety is not widely grown by smallholders and is unavailable in the HIMACUL nurseries. This suggests that farmers obtain this variety from sources other than HIMACUL. Regarding the HAES 772 variety, our findings highlight its popularity among farmers in the highland areas of Chikwatula and

Tithandizane cooperatives. The preference for this variety is attributed to its quick maturity and extended flowering periods, which result in higher yields [57].

In the broader context of countries like Malawi, Kenya, and Mozambique, where smallholder macadamia production is prevalent, our findings underscore the critical importance of involving smallholders in the introduction and breeding of macadamia varieties. This is because while commercial plantations may rely on well-researched varieties tailored to their specific production areas, the diverse microclimates within a country like Malawi emphasize the need for a cautious approach to varietal recommendations especially for smallholder communities.

## Macadamia production challenges

Table 6 presents the challenges faced by smallholder macadamia producers in Malawi. We have identified that insect pests and diseases, especially stinkbugs and nut borers, are the most prominent limitation affecting smallholder macadamia productivity in the country. This is consistent with Evans [8], who reported that stinkbugs and nut borers are the principal economic pests that suppress nut-in-shell yields and volumes of sellable kernel, resulting in a 5% loss of NIS yield (388 tonnes or $1.6 million) and 8% of the kernel (188 tonnes or $2.9 million) in Malawi. Taylor et al. [58] found that stinkbugs and nut borers can cause economic losses of 40–70%, with farmers in the present study reporting similar economic losses. Despite limited pest management training, farmers reported implementing various measures, such as field sanitation and using organic pesticides like Neem and Nkhadze leaves, to reduce pest damage.

The third significant challenge encountered by smallholder macadamia farmers is limited market access. Our research findings highlight that nut prices are primarily determined by commercial processors operating without government intervention. This situation leaves farmers with minimal bargaining power and limited options for alternative buyers. As such, farmers reported not getting more profits from their macadamia crop. These results align with a previous study by Toit et al. [36], which identified a pricing monopoly held by a small number of large commercial processors in Malawi's macadamia industry.

Limited market access is the third important limitation that smallholder macadamia farmers face. Our results reveal that commercial processors determine nut prices without government involvement, leaving farmers with no bargaining power and no alternative buyers. These findings concur with Toit et al. [36], who reported a monopoly in macadamia nut pricing by a few large commercial processors in Malawi. Nonetheless, the establishment of the Malawi Macadamia Association (MMA), coupled with the inclusion of a smallholder farmer representation, is anticipated to foster greater collaboration between smallholder farmers and processors, potentially leading to improved market access and fairer pricing mechanisms.

The study findings have also shown that wind significantly threatens macadamia productivity in some of the study areas. Farmers from Chikwatula, Mphaza, Mwanza, and Tithandizane cooperatives have reported the damaging effects of strong winds, particularly Chiperoni winds that occur between May and July. A farmer remarked:

"These winds lead to annual damage of our macadamia trees, resulting in the loss of flowers and premature nut drop. This, in turn, significantly reduces the overall crop yields."

(Male farmer, Mwanza cooperative).

Given the severity of this issue, the implementation of windbreaks tailored to areas prone to strong winds becomes crucial for maintaining the health and productivity of macadamia orchards in these regions.

Inadequate access to agricultural extension services is another significant barrier to smallholder farmers' productivity. There is empirical evidence that extension contact influences the adoption of production-enhancing techniques by farm households [46, 59, 60]. The results of our study indicate that there is a scarcity of macadamia extension service providers within the selected study areas. As a result, only a small number of farmers currently have access to training and information regarding macadamia production. This has negatively impacted productivity, and some farmers have abandoned their trees due to insufficient support in addressing technical challenges. As a result, the Malawi government and other stakeholders must make substantial investments in agricultural extension services to ensure the long-term sustainability of macadamia production.

## Conclusions and implications

This study aimed to provide valuable insights into the factors influencing smallholder farmers' preferences for macadamia varieties and the major challenges affecting macadamia production and productivity. We have identified that smallholder macadamia farmer preferences for varieties are driven by a complex criteria and perceptions. High yield potential, nut quality (weight and size), and extended flowering patterns are the most critical factors driving farmer varietal preferences. This may be true in other parts of the world where smallholder macadamia farming is common such as Kenya and recently Mozambique, Myanmar and Zambia. Our results show that the top five preferred macadamia varieties among smallholder farmers are HAES 660, 800, 791, 816, and 246. Therefore, it is crucial to consider farmer preferences during varietal introductions and breeding processes to help increase adoption rates and promote sustainable smallholder macadamia productivity. Our study has also revealed that smallholder macadamia farmers encounter various production challenges, including insect pests, diseases, limited market accessibility, wind damage, and a lack of adequate agricultural extension services. To address these challenges, improving the provision of quality agricultural extension services is crucial. In conclusion, by considering farmer preferences and providing quality agricultural extension services, policymakers, and stakeholders can significantly boost productivity and foster the sustainable growth of smallholder macadamia farming in Malawi and other parts of the world especially in SSA and Asia. Achieving this goal necessitates collaborative efforts among various stakeholders, including farmer groups, cooperatives, the private sector, and the government.

## Supporting information

**S1 Table. Characteristics of macadamia varieties cultivated in Malawi.**
(DOCX)

## Acknowledgments

We are grateful to the HIMACUL smallholder farmers for their willingness to engage in discussions with us and their warm welcome. We would also like to thank Andrew Emmott and Dr. Will Rawes, Neno Macadamia Trust, for their assistance during fieldwork planning, as well as Prof. Rick Brandenburg, North Carolina State University, and the Feed the Future Innovation Lab for Peanuts for providing moral and academic support to Emmanuel Junior Zuza. However, any mistakes or omissions in this paper are solely our responsibility.

## Author Contributions

**Conceptualization:** Emmanuel Junior Zuza, Andrew Emmott, Ken Mkengala.

**Data curation:** Emmanuel Junior Zuza, Kadmiel Maseyk, Edwin Kenamu.

**Formal analysis:** Emmanuel Junior Zuza, Kadmiel Maseyk, Shonil Bhagwat, Andrew Emmott, Patrick Phiri, Ken Mkengala.

**Funding acquisition:** Yoseph N. Araya, Kadmiel Maseyk, Shonil Bhagwat, Andrew Emmott.

**Investigation:** Andrew Emmott, Edwin Kenamu.

**Methodology:** Emmanuel Junior Zuza, Shonil Bhagwat, Ken Mkengala.

**Project administration:** Yoseph N. Araya.

**Resources:** Yoseph N. Araya, Rick L. Brandenburg, Andrew Emmott.

**Software:** Emmanuel Junior Zuza, Patrick Phiri, Edwin Kenamu.

**Supervision:** Kadmiel Maseyk, Shonil Bhagwat, Rick L. Brandenburg, Will Rawes, Ken Mkengala.

**Validation:** Emmanuel Junior Zuza, Kadmiel Maseyk, Shonil Bhagwat, Andrew Emmott, Patrick Phiri.

**Visualization:** Emmanuel Junior Zuza, Will Rawes.

**Writing – original draft:** Emmanuel Junior Zuza, Ken Mkengala.

**Writing – review & editing:** Yoseph N. Araya, Shonil Bhagwat, Rick L. Brandenburg, Andrew Emmott, Will Rawes, Patrick Phiri, Edwin Kenamu.

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
