## [Decision Letter · Decision Letter 0]

18 Dec 2023

PONE-D-23-33234Farmer preference for macadamia varieties and constraints to production in MalawiPLOS ONE

Dear Dr. Zuza,

Thank you for submitting your manuscript to PLOS ONE. After careful consideration, we feel that it has merit but does not fully meet PLOS ONE’s publication criteria as it currently stands. Therefore, we invite you to submit a revised version of the manuscript that addresses the points raised during the review process.

We look forward to receiving your revised manuscript.

Kind regards,

Nicholas C. Manoukis

Academic Editor

PLOS ONE

“The corresponding author received funding from the Global Challenges Research Fund to pursue their PhD studies at the Open University. Additionally the corresponding author received training from EarthWatch Community Science Camp (NERC-UKRI Grant no. NE/S017437/1).”

“We are grateful to the HIMACUL smallholder farmers for their willingness to engage in discussions with us and their warm welcome. Additionally, we acknowledge the postgraduate early career researcher support given to Emmanuel Junior Zuza as a member of the EarthWatch Community Science Camp (NERC-UKRI Grant no. NE/S017437/1). We would also like to thank Andrew Emmott and Dr. Will Rawes, Neno Macadamia Trust, for their assistance during fieldwork planning, as well as Prof. Rick Brandenburg, North Carolina State University, and the Feed the Future Innovation Lab for Peanuts for providing a travel grant to Emmanuel Junior Zuza in 2022 for further verification of the data in Malawi.  However, any mistakes or omissions in this paper are solely our responsibility.”

“The corresponding author received funding from the Global Challenges Research Fund to pursue their PhD studies at the Open University. Additionally the corresponding author received training from EarthWatch Community Science Camp (NERC-UKRI Grant no. NE/S017437/1).”

5. In the online submission form, you indicated that [The questionnare data is available upon reasonable request. For the focus group discussions the data is in written form and not digitised.].

6. We note that Figure 1 in your submission contain [map/satellite] images which may be copyrighted. All PLOS content is published under the Creative Commons Attribution License (CC BY 4.0), which means that the manuscript, images, and Supporting Information files will be freely available online, and any third party is permitted to access, download, copy, distribute, and use these materials in any way, even commercially, with proper attribution. For these reasons, we cannot publish previously copyrighted maps or satellite images created using proprietary data, such as Google software (Google Maps, Street View, and Earth). For more information, see our copyright guidelines: http://journals.plos.org/plosone/s/licenses-and-copyright.

Additional Editor Comments:

Both reviewers see value in this study, but agree that the paper required major revisions before it can be accepted. Reviewer 1 suggests a complete rewrite; I believe this can be very beneficial if there is a focus on the organization and presentation, even if parts of the current draft are carried over.

Reviewers' comments:

Reviewer's Responses to Questions

**Comments to the Author**

1. Is the manuscript technically sound, and do the data support the conclusions?

Reviewer #1: Yes

Reviewer #2: Yes

2. Has the statistical analysis been performed appropriately and rigorously? 

Reviewer #1: Yes

Reviewer #2: Yes

3. Have the authors made all data underlying the findings in their manuscript fully available?

Reviewer #1: Yes

Reviewer #2: Yes

4. Is the manuscript presented in an intelligible fashion and written in standard English?

Reviewer #1: No

Reviewer #2: Yes

5. Review Comments to the Author

Reviewer #1: This manuscript discusses the study on the farmer preference for macadamia varieties and challenges to macadamia production across several agricultural districts in Malawi.

The study scale (relative to the size of the industry) was sufficient and the approaches in obtaining and analyzing the data were appropriate. Importantly, the information obtained from this study is relevant in informing research, extension and policy priorities important to the macadamia industry in Malawi.

However, I do recommend that the authors re-write the manuscript to make it concise, straightforward and minimize redundancies. Here are some suggestions: 1) only summarizing the key points and refrain from re-mentioning the information already presented on graphs and tables, 2) removing connectives and verbose descriptives, 3) excluding unnecessary introductions to sections and sub-sections, and 4) deleting the quotes from survey answers, etc. The paper as it is currently written and formatted resembles a feature article in a trade magazine or a comprehensive project report than a peer-reviewed research article.

I would recommend major writing revision of this manuscript before considering for acceptance.

Reviewer #2: this study analyzes macadamia nut cultivar traits that drive varietal selection in Malawi. The study seems to have been well conducted, and the results are clear. It would be good if the authors were able to make the work more than a local case study though; what relevance does this work have outside of Malawi, for example.

Specific comments below; see attached PDF for context.

Annotation Summary of PONE-D-23-33234_reviewer.pdf.

Note [page 11]: More stable than what? This is not necessarily true, there were large fluctuation sin macadamia prices a few years ago.

Note [page 16]: what is this symbol?

Strikeout [page 19]: 3) requiresa substantial variation in the dependent and independent outcomes, preferably from a large enough sample size.

Note [page 19]: This the dependent binary (response) variable, is it not? A dummy variable is a factor included as an explanatory one.

Note [page 25]: I believe at least some of these cultivars have updated names, e.g. HAES660 is known as Keaau.

Strikeout [page 25]: In the Mwanza cooperative, farmers expressed the greatest preference for cultivars HAES 800.

Note [page 30]: I am not sure this is correct! If they are farming, they are active. It seems ageist to suggest that being 50 years old precludes one from being within the “active farming years”.

Note [page 31]: Does this refer to Malawi? The sentence is a bit confusing.

Note [page 32]: compared with areas with more seasonal growth patterns?

Note [page 33]: What does generations refer to here? Distinct periods of cultivar development?

Note [page 34]: How does this pattern of variety selection compare with other places in the world? 344 is a popular cultivar in Hawaii, for example.

Note [page 35]: du?

Note [page 37]: Are there lessons for other production areas, or other production systems? Does this work have any bigger-picture contribution?

Note [page 41]: du Toit, i think?

6. PLOS authors have the option to publish the peer review history of their article (what does this mean?). If published, this will include your full peer review and any attached files.

Reviewer #1: No

Reviewer #2: No

---

## [Author Response · Author response to Decision Letter 0]

19 Dec 2023

Responses to Academic Editor

File naming and style requirements:

• This has been addressed, and the manuscript now adheres to PLOS ONE's style requirements, including proper file naming.

Inclusivity in global research questionnaire:

• The complete copy of PLOS' questionnaire on inclusivity in Global research has been included in the revised manuscript.

Role of funders:

• The role of funders in the study has been clearly stated in the new version of the manuscript.

Removal of funding-related text:

• All funding-related text has been removed from the manuscript as recommended.

Data accessibility:

• The questionnaire data has also been made available here uploaded. Focus group discussion data, being sensitive, is available upon request.

Permission for figures:

• All data from OpenTopography is freely available and published under the CC BY 4.0 license.

Responses to Reviewer #1:

Summarizing key points:

• Quotes from focus group discussions have been retained to maintain the integrity of the study and inform policy. However, we have summarised them as suggested. 

• Similar styles have been employed in transdisciplinary papers as well and hence the reason of us doing this.

Connectives and descriptives:

• Redundant connectives and verbose descriptives have been removed as recommended.

Unnecessary introductions:

• Unnecessary introductions to sections and sub-sections have been excluded in the revised manuscript.

Quotes from survey answers:

• Quotes from survey answers have been retained, considering the mixed methods approach used for data collection.

• The reason for this decision is because the qualitative data enriches the manuscript and provides an importance of including community knowledge in research which is currently being advocated for. Some examples of other papers that have done the same include:

• https://journals.plos.org/plosone/article?id=10.1371/journal.pone.0290877

• https://journals.plos.org/plosone/article?id=10.1371/journal.pone.0272126

• https://rmets.onlinelibrary.wiley.com/doi/pdf/10.1002/cli2.55

• https://journals.plos.org/climate/article?id=10.1371/journal.pclm.0000082

• https://journals.ametsoc.org/view/journals/wcas/14/2/WCAS-D-21-0075.1.xml

Responses to Reviewer #2:

Relevance outside of Malawi:

• Policy recommendations have been included to emphasize the broader relevance beyond Malawi.

• For example, we advocate for informed breeding programs that align with farmer preferences to promote greater adoption of macadamia varieties (the reason for this is because majority of macadamia breeding programs do not consider smallholder farmer preferences rather commercial estates).

Clarification of macadamia prices:

• The statement regarding macadamia prices has been revised to provide clarity on the crop's returns.

• It now reads: Because the crop's returns are higher (~$14 to 15 kg-1) than other cash crops, the country's macadamia industry is rapidly expanding, with farms previously used for coffee, tea, and tobacco production being converted or diversified to macadamia production [10,11].

Explanation of symbol :

• The symbol "For all" has been explained for better clarity in the manuscript.

• Note for all varieties in set with elements j, k, the farmer chooses a variety that gives them the most utility/profit when they compare j and k.

Correction of writing error:

• The writing error of requiresa has been changed to: requires a substantial variation in the dependent and independent outcomes, preferably from a large enough sample size

• The writing error regarding the dependent binary variable has been rectified.

Verification of cultivar names:

• The cultivar names have been verified, and the correct information has been incorporated into the manuscript.

• Note Keaau is an old name of HAES 660 base on the following publication (here).

Amendment of varietal preferences in Mwanza cooperative:

• Preferences for varieties in Mwanza have been amended to accurately reflect the data i.e. HAES 741 and 800 are the most preferred (24%).

Clarification on "Active Farming Years":

• The explanation of considering 50 years as less active in farming in the context of Malawi has been provided.

• In Malawi once you are at the age of 50 and above you are considered less or not active in farming. The reason being the use of hand-held implements and at that age the amount of work you can do is limited and hence the expression.

Does this refer to Malawi? The sentence is a bit confusing:

• The sentence referring to Malawian macadamia processors has been rephrased for clarity.

Compared with areas with more seasonal growth patterns?

• Navigating this question proves challenging as our available data lacks support for the underlying assumption. Consequently, we opted to narrow our focus to examine the impact of topography on the flowering patterns of HAES 791 in comparison to HAES 660. The manuscript underscores that certain varieties exhibit extended flowering periods, while others undergo flowering for only 2 to 3 months, contingent upon the specific geographical area.

What does generations refer to here? Distinct periods of cultivar development?

• The answer is Yes. We have rectified this to avoid the confusion. 

How does this pattern of variety selection compare with other places globally? For instance, 344 is a popular cultivar in Hawaii:

• The manuscript now includes information on how variety selection differs between smallholder and commercial farmers in various regions.

• It is essential to note that there are significant disparities in how variety selection is conducted between smallholder and commercial producers of macadamia. Among smallholders in Malawi and Kenya, individuals typically lack influence over the varieties they cultivate and are not actively engaged in breeding and introduction programs. They rely on recommendations from agronomists associated with commercial estates, often situated in different production regions. The high significance of our manuscript lies in emphasizing the crucial inclusion of smallholders in varietal breeding and introduction research. 

• Regarding HAES 344, its popularity in Hawaii stems from its specialized breeding for the region. Notably, it is currently being replaced by newer varieties. Additionally, this particular variety is predominantly cultivated in Malawi and not in South Africa (RSA) and Kenya among commercial producers.

Bigger-picture contribution:

• The manuscript now explicitly states the lessons learned and the broader contributions to varietal breeding research, especially in areas with smallholder macadamia production.

• Our research holds relevance for various production areas and systems. It marks the initial study focusing on smallholder farmer macadamia varietal preferences. In regions where smallholder macadamia production is prevalent, crucial lessons have been gleaned. For instance, before introducing or breeding macadamia varieties, it is imperative to involve smallholders in the research process. 

• An illustrative example is found in Mozambique, where smallholders, encouraged to use the Beaumont variety as a rootstock, encountered significant challenges due to unsuitable conditions. This highlights the potential pitfalls of adopting varieties without considering local conditions. 

• Furthermore, while estate producers often use well-researched varieties tailored to specific regions, the diverse microclimates within a country, such as Malawi, necessitate caution. Our study reveals regional variations, such as HAES 791's longer flowering period in specific parts of the country, emphasizing the importance of localized research. 

• These findings have broader applicability beyond Malawi, offering insights for macadamia cultivation in diverse regions worldwide. Additional sentences have been incorporated to underscore these points as recommended.

Citation software error:

• The citation software error regarding "du Toit" has been rectified.

We believe that these revisions address the concerns raised by the editor and reviewers comprehensively. We appreciate the time and effort invested in reviewing our manuscript, and we are confident that the changes made enhance the overall quality of the submission.

---

## [Decision Letter · Decision Letter 1]

11 Jan 2024

Farmer preference for macadamia varieties and constraints to production in Malawi

PONE-D-23-33234R1

Dear Dr. Zuza,

We’re pleased to inform you that your manuscript has been judged scientifically suitable for publication and will be formally accepted for publication once it meets all outstanding technical requirements.

Kind regards,

Nicholas C. Manoukis

Academic Editor

PLOS ONE

Additional Editor Comments:

Thank you for doing a nice job addressing comments and suggestions, and congratulations on completing a nice study.

Reviewers' comments:

Reviewer's Responses to Questions

**Comments to the Author**

1. If the authors have adequately addressed your comments raised in a previous round of review and you feel that this manuscript is now acceptable for publication, you may indicate that here to bypass the “Comments to the Author” section, enter your conflict of interest statement in the “Confidential to Editor” section, and submit your "Accept" recommendation.

Reviewer #1: All comments have been addressed

Reviewer #2: All comments have been addressed

2. Is the manuscript technically sound, and do the data support the conclusions?

Reviewer #1: Yes

Reviewer #2: (No Response)

3. Has the statistical analysis been performed appropriately and rigorously? 

Reviewer #1: Yes

Reviewer #2: (No Response)

4. Have the authors made all data underlying the findings in their manuscript fully available?

Reviewer #1: Yes

Reviewer #2: (No Response)

5. Is the manuscript presented in an intelligible fashion and written in standard English?

Reviewer #1: Yes

Reviewer #2: (No Response)

6. Review Comments to the Author

Reviewer #1: The authors were able to adequately address the comments and changes I recommended. As well, they provided good justifications for those that they retained.

Reviewer #2: (No Response)

7. PLOS authors have the option to publish the peer review history of their article (what does this mean?). If published, this will include your full peer review and any attached files.

Reviewer #1: No

Reviewer #2: **Yes: **Mark G Wright

---

## [Editor Report · Acceptance letter]

12 Feb 2024

PONE-D-23-33234R1 

PLOS ONE

Dear Dr. Zuza, 

I'm pleased to inform you that your manuscript has been deemed suitable for publication in PLOS ONE. Congratulations! Your manuscript is now being handed over to our production team.

Kind regards, 

on behalf of

Dr. Nicholas C. Manoukis 

Academic Editor

PLOS ONE